# Practical experience and framework for sensitivity analysis of hydrological models: six methods, three models, three criteria

Anqi Wang<sup>1</sup> and Dimitri P. Solomatine<sup>2,3,4</sup>

<sup>1</sup> College of Hydrology and Water Resources, Hohai University, NO.1 Xikang Road, Nanjing, 210098, China

<sup>2</sup> Chair of Hydroinformatics, IHE Delft Institute for Water Education, Westvest 7, Delft, 2611AX, The Netherlands

<sup>3</sup> Water Problems Institute, Russian Academy of Sciences, Leninsky prospekt 14, Moscow, 119991, Russia

<sup>4</sup> Water Resources Section, Delft University of Technology, Postbus 5, Delft, 2600AA, The Netherlands

Correspondence to: Anqi Wang (wanganqi0718@163.com)

Abstract. Sensitivity Analysis (SA) and Uncertainty Analysis (UA) are important steps for better understanding and evaluation

- of hydrological models. The aim of this paper is to briefly review main classes of SA methods, and to presents the results of the practical comparative analysis of applying them. Six different global SA methods: Sobol, eFAST, Morris, LH-OAT, RSA and PAWN are tested on three conceptual rainfall-runoff models with varying complexity: (GR4J, Hymod and HBV) applied to the case study of Bagmati basin (Nepal), and also initially tested on the case of Dapoling-Wangjiaba catchment in China. The methods are compared with respect to effectiveness, efficiency and convergence. A practical framework of selecting and
- using the SA methods is presented. The result shows that, first of all, all the six SA methods are effective. Morris and LH-OAT methods are the most efficient methods in computing SI and ranking. eFAST performs better than Sobol, thus can be seen as its viable alternative for Sobol. PAWN and RSA methods have issues of instability which we think are due to the ways CDFs are built, and using Kolmogorov-Smirnov statistics to compute Sensitivity Indices. All the methods require sufficient number of runs to reach convergence. Difference in efficiency of different methods is an inevitable consequence of the
- differences in the underlying principles. For SA of hydrological models, it is recommended to apply the presented practical framework assuming the use of several methods, and to explicitly take into account the constraints of effectiveness, efficiency (including convergence), ease of use, as well as availability of software.

# **1** Introduction

Hydrological models are widely used to simulate natural phenomena, mainly for the purpose of generating forecasts.
Deterministic forecasts inevitably raise the issue of its uncertainty. This uncertainty mainly comes from the error of gathering input data, e.g. rainfall and evapotranspiration, parameters of the model and the model structure itself. Nowadays, the interests to Uncertainty Analysis (UA) methods and procedures have grown considerably. The study of the UA will not only improve the credibility of the model itself but also be conductive to decision making under uncertainty.

There are various definitions of UA proposed by different researchers (Cacuci, 2005; Pappenberger and Beven, 2006; Tong, 2006; Saltelli et al, 2008; Bastin et al, 2013). UA gives a qualitative or quantitative assessment of the uncertainty in the model results. The results can be qualitatively expressed in a graph showing the spread an ensemble of values or a distribution, as probabilistic flood maps, etc.

5

Due to the complexity and non-linear nature of hydrological models, it is hard to use analytical methods to study the uncertainty of hydrological models. Therefore, non-intrusive, sampling-based methods are commonly used, generally referred to as Monte Carlo Simulation (MCS), which can be seen as the simulation of a system that encloses stochastic or uncertain components. It can be easily implemented, model independent and dimension independent.

The Monte Carlo method can be expressed as "the use of random sampling as a tool to produce observations on which 10 statistical inference can be performed to extract information about a system" (Lemieux, 2009). The main steps of MCS are firstly generating n samples of input X over the input variable space. The input variables can be external model inputs, initial model conditions or model parameters. For each such realisation, simulation of the model Y=f(X) is carried out, to obtain n sets of output (could be either time series or single value), which statistics are analysed.

There is another concept used to analyse the impact of uncertainties on modelling results, Sensitivity Analysis (SA), which is ideologically close to UA. It can be defined as the study of "how the uncertainty in the output of a model (numerical 15 or otherwise) can be apportioned to different sources of uncertainty in the model input" (Saltelli et al., 2008). (One may notice that this definition is not comprehensive, since uncertainty not only comes from model inputs but also from parameters, so for this reason, we will use the term "factor" instead of "model input".) The main aim here is to identify the degree with which changes in various factors (manifesting the corresponding uncertainty) influence a change in model output. SA should be seen

as a standard step in any modelling study, and there is plenty of literature on SA published during the last 40-50 years, but still 20 various updates and improvements of SA techniques are proposed regularly (see e.g. Razavi and Gupta, 2016a, 2016b; Pianosi and Wagener, 2015).

SA is often implemented before model parameterization (calibration). On one hand, for conceptual rainfall-runoff models, the parameters cannot be gathered from field measurement, implementing, and SA can help to find out the most influential 25 parameters to reduce the cost of calibration time. On the other hand, for distributed hydrological models, whose parameters can be gathered from the field, SA can help to target the most important parameters, on which more resources can be put to ensure their higher accuracy. (It should be noted that there is a certain danger and even a methodological flaw in conducting SA of parameters before model calibration: it is not yet really known what is the optimal parameter vector, and hence it is possible that sensitivity is investigated considering non-feasible parameters values. So it would be advisable to carry out at least some initial calibration before turning to SA.)

30

SA is conducive to UA, and the main difference between their aims lies in that SA tries to explicitly apportion the uncertainty of the output to the different factors. Therefore, SA can help to target the sources of the model output uncertainty

due to that in inputs or parameters, whereas UA provides a more general and often more detailed and rigorous account of model uncertainty.

Saltelli et al. (2008) formulates the three main specific purposes of SA:

- Factor Prioritization (FP): ranking the factor in terms of their relative sensitivity;
- Factor Fixing (FF), or screening: determining the factors are influential or not to the output uncertainty;
  - Factor Mapping (FM): given specific output values or ranges, locating the regions in the factor space that produces them. In this study, we only focus on ranking and screening.

SA is typically categorized into Local Sensitivity Analysis (LSA) and Global Sensitivity Analysis (GSA). LSA concentrates on the sensitivity of factors at particular points in the factor space, for example, around the vector of the calibrated

parameters. GSA, on the other hand, assesses the sensitivity of the factor through the whole factor space. By design, LSA is simpler and faster.

A simplest expression of local sensitivity is the first-order partial derivatives of output to the factors. Define a model y = f(x), where y is the output of the model; x is factor of the model. The sensitivity of the factor (S) is defined as:

$$S_i = \frac{\Delta y_i}{\Delta x_i} \tag{1}$$

where i is the i-th factor of the model. (Note, that in quite many studies instead of model output y the model error is used, e.g. Root Mean Squared Error or Mean Absolute Error.) Higher value of Si indicates higher sensitivity of the factor. Such measure of sensitivity is often called Sensitivity Index (SI). Figure 1 shows the expression of sensitivity of a model with two parameters (factors).

If we randomly sample several points in the whole parameter space, and obtain Si for each sample point. After that we can aggregate the results (e.g. calculating the mean value of these Si), assessing thus the global sensitivity of the model.

Global Sensitivity Analysis methods can be classified into Generalised Sensitivity Analysis method, variance-based methods, GLS (globally aggregated measure of local sensitivities) methods, density-based methods and meta-modelling methods. Different methods arew based on different theories and principles, and as a result, have different efficiencies. Which method is the best to use is always an issue to discuss in the field. There are various studies comparing different SA methods.

- In the study of Tang et al. (2007), four SA methods have been analysed and compared on SAC-SMA coupled with SNOW-17. The results of the study show that the choice of SA methods has great impacts on the parameter sensitivity of the model. Pappenberger et al. (2008) tested five SA methods on a flood inundation model (HEC-RAS). It is demonstrated that different methods result in different ranking of factors, thus solid conclusions about the sensitivity of the factors are impossible to draw. Gan et al. (2014) have evaluated the effectiveness and efficiency of ten widely used SA methods on SAC-SMA model. The
- result demonstrates qualitative SA methods are more efficient than quantitative SA methods, whereas quantitative SA methods are more robust and accurate. Song (2015), Razavi and Gupta (2015) and Pianosi et al. (2016) gave systematic reviews of SA concepts, methods and framework respectively. Suggestions on how to choose SA methods are provided. However, these



suggestions are only made based on dissimilar studies and literature reviews, a comprehensive comparison of SA methods applied to one case study is lacked in their studies.

With respect to sample-based SA methods, the coverage of the factor space is the key point in SA accuracy: with small samples the SA results are imprecise. In other words, there is also uncertainty in SA (in fact, the same can be said about UA). In order to deal with this issue, convergence of the Sensitivity Indices should be studied.

In spite of many reviews and comparison of SA methods that have been carried out, there are not too many studies that investigate the convergence and uncertainty of the SA results. Yang (2011) assessed the convergence of Sensitivity Indices for five different Global Sensitivity Analysis methods using Central Limit Theory (CLT) and bootstrap techniques. In her study, the estimates of mean and Confidence Interval (CI) are plotted against increasing base sample size for each method. Once there is no significant fluctuation in the values, the convergence is reached. Sarrazin et al. (2016) proposed a methodology to study the convergence of Sensitivity Indices, ranking and screening. They have defined quantitative criteria for the convergence of Sensitivity Indices, ranking and screening, and tested the methodology on the three widely-used GSA methods

applied to three hydrological models.

Yet another aspect worth attention is the choice of SA method(s). Most of the studies concerning SA could not draw firm 15 conclusions about how to choose the best SA method (and this is understandable since there are many ways to define what is the "best" one). Also, the uncertainty in SA is not investigated much.

The first objective of the study is to test and compare the widely used classic SA methods as well as the SA methods developed recently (e.g., PAWN, Pianosi and Wagener, 2015a) in the aspects of efficiency, effectiveness and convergence. The second objective is to give suggestions on how to choose SA methods for various hydrological (or hydraulic models)

based on their computational cost, robustness and easiness of implementation. The third objective of the study is to formulate a practical framework of sensitivity and uncertainty analysis of hydrological models, thus contributing to and complementing the guidelines published earlier (e.g. Saltelli et al., 2008; Baroni and Tarantola, 2014; Song et al., 2015; Pianosi et al., 2016). Each individual SA study has its specifics so it is hardly possible to have a unified framework or procedure that would fit all possible requirements. Each researcher or practitioner would have a choice of various approaches, principles and components

```
to combine and follow in SA.
```

The structure of the paper is as follows. Section 2 gives detailed introduction and description of Global Sensitivity Analysis methods. Section 3 presents the methodology and case study of this study to evaluate GSA methods. The results of the study are shown in Sect. 4 and followed by discussions in Sect. 5. Finally, conclusions are drawn in Sect. 6.

#### 2 Global Sensitivity Analysis methods

This section is in no way a detailed presentation of the methods, but rather a brief introduction to the techniques compared in this study. For comprehensive reviews, please refer to Song et al. (2015), and Pianosi and Wagener (2015), and for a relatively recent interesting insight into the SA problem to Razavi and Gupta (2015, 2016a, 2016b).

#### 2.1 Classification of GSA methods 5

#### 2.1.1 Generalised (Regionalised) Sensitivity Analysis method

Generalized Sensitivity Analysis method (also referred to as Regionalised SA) has gained popularity in environmental and water-related research in the end of 1970s, especially after the papers by Spear and Hornberger (1980, 1981); to some extent Whitehead and Young (1979), and it is also worth checking the earlier work by Spear (1970). This approach was positioned

- as a Monte Carlo framework used for "probabilistic calibration" which aimed at finding regions in parameter space leading 10 sets of "behavioural" (good) and "non-behavioural" (bad) models (which point at the regions of critical uncertainty), rather than aiming at finding one "best" model. Simulation results are split into these two groups based on their performance (e.g. model error), the Cumulative Distribution Function (CDF) of each factor is generated for each group, and their difference is analysed. Typically, the Kolmogorov-Smirnov statistic (Massey Jr, 1951) is used to compute the discrepancy between these
- CDFs. GSA allows for identifying the regions of the model parameter space in which parameters have the significant effect 15 on the model behaviour. One can see also that GSA, being based on the Mote Carlo framework and using statistical analysis of outputs, can be also seen as a representative of UA.

One of the drawbacks of RSA is that the results are influenced by the selection of different thresholds and so this undermines its objectivity. To resolve this problem, Wagener et al. (2001) presented an extension of this method. The parameter sets are grouped into ten groups instead of two, based on the model performance. They are sorted from best to worst, 20 in which the first group produces the best 10% results (e.g. the results with least 10% model error), the second group produces the best 10%-20% results and so on. Empirical CDFs of the parameters are also plotted for each group, if the curves are concentrated or overlapped, the parameter are not sensitive, and vice versa. For detailed description and implementation of the method, please refer to Jakeman et al. (1990) and Wagener et al. (2001).

#### 25 2.1.2 Variance-based methods

Variance-based methods are today the most popular approaches for SA. The underlying assumption of variance-based methods is that the sensitivity can be measured by the contribution of the factor's variance (the contribution of the factor itself, or interactions with two or more factors) to the variance of the output. The biggest advantage of a variance-based method is that it can compute the main effect and higher-order effect of factors respectively, and make it distinguishable which factor have high influence on the output by its own, and which factor have high interaction with others.


It is normally unrealistic to analytically compute the Sensitivity Index because of the complexity of hydrological models. Instead, Sobol' proposed an efficient sample-based approach to compute first and total-order Sensitivity Indices - the called Sobol' method - which is perhaps the most popular variance-based SA method (Sobol', 1993). A detailed description of the method and its implementation can be also found in Saltelli et al. (2008).

- Though the result of Sobol' method is robust, often considered as benchmark run for study, it is computationally expensive, requiring large number of base samples. Another popular approach to numerically compute variance-based Sensitivity Indices is the Fourier Amplitude Sensitivity Test (FAST), presented by Cukier et al. (1973). The key idea of FAST is applying the ergodic theorem to transform the *n*-dimension integral to one-dimension integral. Saltelli and Bolado (1998) provide a detailed description of principles and procedures for implementation of the method. One of the drawbacks of FAST method is that it
- can only compute the main effect. However, an improved version of FAST, which is extended FAST (eFAST, Saltelli, et al.,
   1999), can compute first and total order Sensitivity Indices.

#### 2.1.3 Globally aggregated measure of local sensitivities (GLS) method

As mentioned in sect. 1, the globally aggregated measure of local sensitivities methods use average value of SA measures (e.g. first-order derivative) at each local sample points in the factor space as Sensitivity Index for each factor.

- Morris (1991) proposed an approach which he referred to as Elementary Effects Test (EET) to compute the sensitivity. It is also called Morris Screening method. Its modification was proposed by Campolongo et al. (2007). Its principle concept is to use the mean and standard deviations of the gradients of each sample as the measure of the overall effect and interaction effect of each factor across the p level factor space. Morris Screening is a simple but effective method, widely used for screening in hydrological modelling. A more detailed description of the method can be found in Saltelli et al. (2008).
- Since sampling is time-consuming, it is reasonable to use economical techniques for it, and e.g., van Griensven et al. (2006) employed Latin Hypercube Sampling, followed by assessments of the local error derivatives at each point "one at a time" (OAT), which they named LH-OAT method. The Sensitivity Index of each factor is obtained by averaging the derivatives of all perturbed samples.

All GLS methods conceptually are quite simple and their reported implementations typically do not require large number of runs. However, Razavi and Gupta (2015) have pointed out that they may suffer from scale issue, that is, the selection of the step size may influence the results due to the complexity of response surface of the model.

#### 2.4 Density-based methods

Both GLS methods and variance-based methods are moment-dependent approaches, which use the first moment (first-order derivatives) or the second moment (variance) to compute Sensitivity Indices. The density-based methods do more, and explore

PDFs or CDFs of the output. Sensitivity is measured by the comparison of unconditional PDF derived from purely random samples and conditional PDF derived when prescribing one factor. Entropy-based sensitivity measures (Park and Ahn, 1994;


Krykacz-Hausmann, 2001; Liu et al., 2006) and the  $\delta$ -sensitivity measure (Borgonovo, 2007; Plischke et al., 2013) are implementations of this concept.

The production of empirical PDFs is a crucial step in most of the density-based methods. However, the derivation of empirical PDFs is either too simple, so that the results may not be accurate, or too complex to implement. Recently, Pianosi and Wagener (2015a) proposed a novel method called PAWN that partly overcomes this difficulty. The key idea of PAWN method is to compare the unconditional CDF of output with conditional CDFs of output which prescribe one parameter at a fixed value (the conditioning value) while others vary randomly.

#### 2.1.5 Use of meta-modelling to reduce running times

Sampling used in SA requires considerable computational time, for complex models prohibitively long. The basic idea of meta-modelling is to substitute the original model (and hence its response function linking factors and model output) with a simpler function or a model. This substitution is typically done by using statistical or machine (statistical) learning techniques, and employing methods of the so-called experimental design for generating data by the model runs to be used for training the meta-model. SA is carried out using the meta-model, and for this mostly variance-based method is used.

Techniques used for this purpose include Radial-basis function network (RBF, Broomhead and Lowe, 1988), multivariate adaptive regression splines (MARS, Friedman, 1991), support vector machine (SVM, Cortes and Vapnik, 1995), Gaussian processes (GP, Rasmussen, 2004) and treed Gaussian processes (TGP, Gramacy and Lee, 2008). The advantage of metamodelling is that by simplification of the original complex model, the overall running time is considerably decreased; the trade-off is a possible loss of accuracy.

#### 3. Methodology and Experimental set-up

# 20 3.1 Methodology for evaluating SA methods

#### 3.1.1 What aspects do we evaluate

Different SA methods have different concepts and principles behind them, and, accordingly, the Sensitivity Indices may have different meaning and metrics. However, it would be logical to try to follow the general principles behind any method for a model (method) evaluation, i.e. effectiveness and efficiency. The evaluation of SA methods' *effectiveness* is aimed at finding

- 25 out whether the relative Sensitivity Indices, ranking and screening of parameters have sense and indeed can be used in SA. *Efficiency* of SA methods is assessed by how fast (in terms of computational time) they provide the result: the lesser number of model runs is required, the more efficient the method is. Therefore, the evaluation of SA methods efficiency is to figure out the minimum number of runs required for each SA method to get satisfactory results and it is not always clear and explicitly defined what "satisfactory" actually means. Due to the fact that sampling is employed, there is always uncertainty in the SA
- 30 results, and the values of the Sensitivity Indices calculated depend on the sample size. In order to take into account the

uncertainty nature of SA results, the *convergence* of the SA results should be studied, and this forms the last aspect of the evaluation of SA methods.

# 3.1.2 Evaluation of effectiveness

The result of SA is not an absolute one and nobody can say what is the "correct answer". Unlike assessing the accuracy of a

- 5 hydrological model, which can be compared with the observation values, for sensitivity there are no 'observations' to be compared with. To start somewhere, we will initially randomly sample a large number (say, 10,000) parameter (factor) vectors and run the model for each of them. The RMSE of the model output will be plotted against parameter values as a scatter plot which will provide a rough image of the sensitivity of each parameter. The preliminary assessment of the sensitivities of each parameter will be treated as a reference. Then all considered SA methods will be run, and their results will be compared with
- 10 the reference to assess their performance. Effectiveness will be evaluated on the three aspects: Sensitivity Indices values, ranking and screening.

We realize that constructing a reference this way provides quite a rough estimation of sensitivity, and this is an inevitable limitation. Therefore, the results of all the methods will be taken into account, compared and analysed to see the differences and similarities between them and not only with the reference.

# 15 **3.1.3 Evaluation of efficiency**

For each method, one benchmark test will be run with a considerable size of the base sample set of 10,000. Different base sample sizes will be set for each SA method, to be compared with the results of its benchmark run. From the results, the minimum base sample size will be found for each SA method to ensure the effective results in terms of Sensitivity Indices stability and factors ranking.

#### 20 3.1.4 Evaluation of convergence

Convergence of Sensitivity Indices will be analysed by calculating 95% confidence intervals, mean and variance for various sample sizes. To increase the confidence of estimates, bootstrapping (see e.g. Efron and Tibshirani, 1986) will be used as well. The following procedure will be employed (adapted from Yang, 2011):

- 1. Generate N samples of parameters as the base sample set.
- 25 2. The N base samples are re-sampled B times with replacement, and for each replica, the Sensitivity Indices are computed, producing B Sensitivity Indices to construct the distribution of them.

From this sampling distribution, statistics of the Sensitivity Indices distribution is calculated to quantify uncertainty.

#### 3.2 Case study

The presented methods have been tested on two case studies: Dapoling-Wangjiaba catchment in China, and the Bagmati catchment located in central Nepal. Due to data limitations issues not all experiments with the first case have been finalised, so it is not reported here, and is left for the future publications.

- 5 Bagmati catchment covers an area of approximately 3700 km<sup>2</sup> (see Fig. 2). The altitude of the region varies from 2913 m in the Kathmandu Valley, to Terai Plain, where it reaches the Ganges River, in India with an altitude of 57 m. The Bagmati River has an extension of about 195 km, flowing from Shivapuri to the Ganges River in the south. In this study, focus is put on the part of the basin that drains to the Pandheradobhan station, with an area of 2900 km<sup>2</sup> and river length of 134 km.
- In this study, daily precipitation and air temperature from Kathmandu, Hariharpurgadhi, and Daman station and daily discharge in Pandheradobhan station from 1 March 1991 to 31 December 1995 are used. The daily average precipitation was assessed using Theissen polygon method and the potential evapotranspiration is calculated by the modified Penman method recommended by the Food and Agriculture Organization- FAO (Allen, 1998). The hydrograph is shown in Fig. 3.

#### 3.3 Test model

The SA will be tested on three conceptual rainfall-runoff models: GR4J, Hymod and HBV, with increasing complexity 15 and the parameters number.

The modèle du Génie Rural (Agricultural Engineering Model) à 4 paramètres Journalier (4 parameters Daily, GR4J) was developed by Perrin et al. (2003). It uses daily precipitation and evapotranspiration as input to simulate the runoff discharge. The model structure assumes that after neutralization of precipitation by evapotranspiration, a portion of net rainfall goes to production store, where percolation takes place. The leakage flow, together with the remaining part of the net rainfall, go to

routing store, where they are split into two parts and routed by two unit hydrographs. After exchanging with groundwater, the total runoff is generated by adding these two parts. The four parameters with their meaning and ranges are shown in Table 1.

The Hymod model, first introduced by Boyle (2001) and presented in Wagener et al. (2001) has been used quite widely for rainfall-runoff modelling because of its simplicity. It consists of a simple rainfall excess model with two parameters and a routing module with three parameters. In the rainfall excess model, the soil moisture storage capacity is assumed to be variable,

described by a distribution function. The routing module contains two sets of parallel linear reservoirs. Three identical linear reservoirs account for the fast runoff component and a single linear reservoir accounts for the slow runoff component. The name, meaning and ranges of the parameters are shown in Table 2.

The HBV (Hidrologiska Bryåns Vattenbalansaldevning) model is a conceptual rainfall-runoff model widely used in Europe. It was developed by the Swedish Meteorological and Hydrological Institute (Bergström, 1976) and then promoted by

Lindström et al. (1997) to become the HBV-96 model. In this study, a simplified version of the HBV-96 model is used. It consists of the three main modules, which is characterized as tank respectively, with 13 parameters: four of the parameters are related to snow accumulation and melt module, four with soil moisture accounting module and five with river routing and

response module. For river routing and response module, two runoff reservoirs are included. The upper non-linear reservoir accounts for the quick flow and the lower linear reservoir accounts for the base flow. Since there is little snowfall in the applied case study, the snow accumulation and melt module are excluded, so only nine parameters will be analysed. The name, meaning and ranges of the parameters are shown in Table 3.

# 5 3.4 Experimental set-up

The experimental set-up is presented in Table 4. The evaluation is done on six SA methods: Sobol, eFAST, Morris, LH-OAT, RSA and PAWN. All software is implemented in MATLAB. For Sobol method, eFAST. Morris screening, RSA and LH-OAT, the codes are constructed by the first author. For PAWN method, the codes from the SAFE toolbox (Pianosi et al., 2015b) are used. In the present study we follow a widely adopted approach when instead of studying the sensitivity of the model directly,

the sensitivity of the model error (deviation from observations) is analysed instead. For the model error, we use the Root Mean Squared Error ( $E_{RMSE}$ ):

$$E_{RMSE} = \sqrt{\frac{1}{n} \sum_{t=1}^{n} (Q_{sim}^{t} - Q_{obs}^{t})^{2}}$$
(2)

where  $Q_{sim}^t$  is the simulated model output at time step *t*,  $Q_{obs}^t$  is the observation value at time step *t*, *n* is the number of time steps. To avoid the influences of model initial states, the first three months (90 time steps) are excluded when computing  $E_{RMSE}$ .

Due to the characteristics of FAST sampling in eFast method, bootstrapping resample is not applicable, evaluation of convergence will not be done for eFAST method. The resample size for other SA methods for evaluation is 100.

#### 4. Results


#### 4.1 Preliminary assessment of sensitivity

The model was run 10,000 times; the scatter plots of the  $E_{RMSE}$  against parameters for the three models are shown in Fig. 4-6. From the scatter plot, the relative sensitivity of the parameter can be seen from the randomness of its distribution (i.e. proximity to the uniform distribution). The more randomly the RMSEs are distributed, the less sensitive the parameter is.

In the GR4J model (Fig. 4) X4 is shown to be the most influential parameter, followed by X3 and X2, while X1 seems to be of little influence. X4 is the time when the ordinate peak of flood hydrograph is created, which actually determines the

shape of the hydrograph, so it is no surprise it appears to the most influential parameter in the model. X1 is the storage of rainfall in the soil surface, which does not affect the routing process too much, thus it is the least sensitive parameter.

In Hymod model (Fig. 5), ALFA, RS and RF have high influence on RMSE. SM and BETA, however, seem to be noninfluential. This is understandable, because ALFA, RS and RF controls the fast and slow pathway in flow-routing module