# Peer review of "Practical experience and framework for sensitivity analysis of hydrological models: six methods, three models, three criteria"

_Hydrology and Earth System Sciences, 2018_

## Referee Comment (RC1) · K. J. Beven (Referee) · 3 Mar 2018

I am afraid I gave up on this paper (after making quite a lot of comments in the manuscript) at the point where Figures 4-6 are introduced and demonstrate the irrelevance of sensitivity analysis in the chosen case study. Almost certainly in these cases the performance of the model has more to do with uncertainty in the input and output data, that is totally neglected, than the factors included in the analysis. This is indicative of the apparently naïve way the issues associated with sensitivity analysis are presented in the introductory sections which can only be described as poorly presented. In particular, there is no real recognition of the potential for complexity of

surfaces with sometimes rapidly changing covariation, including changes of sign, of factors in producing the outputs (that can be concealed in plots such as Fig 4) - yet such behaviour is common for real model applications. Also, despite the discussions of the last 30 years, the authors still seem (surprisingly?) to believe in the possibility of an optimum calibrated model.

The authors recognise that nearly all past intercomparisons of SA methods have suggested that different methods give different results, and that the same method might give different results when used with different outputs. So it is here too. This is not therefore unexpected, so where is the value in this paper, or in continuing to explore further SA methods as they suggest. Are the results really ever used to decide parameters "on which more resources can be put to ensure their higher accuracy". How would you actually do this for the conceptual models used in the paper, when it is effective values of model parameters that are needed to give good predictions? That would be a much more interesting paper. As it is I cannot suggest that this paper is suitable for publication.

Keith Beven

Please also note the supplement to this comment:
https://www.hydrol-earth-syst-sci-discuss.net/hess-2018-78/hess-2018-78-RC1-supplement.pdf

—————————————

[Figure]

**Supplement:**

[revised manuscript text omitted]

which is more important in determining flows, while SM and BETA only account for soil moisture routine which is less important.

In Fig. 6, it can be seen that in HBV model, MAXBAS is obviously the most influential parameter; FC, ALFA, BETA and PERC also shows certain degree of sensitivity; LP, K, K4 and CFLUX are non-influential. The reason is that MAXBAS is the (routing) transfer function parameter which controls the shape of the hydrograph.

**4.2 Effectiveness**

Figure 7 shows the results of benchmark runs of each method for three models, and one may see the following:

1) all the methods identify the same set of sensitive parameters (X3 and X4 for GR4J, ALFA, RS and RF for Hymod, MAXBAS for HBV);

2) for less influential or non-influential parameters, different methods show relatively large discrepancy in results;

3) the results of Sobol and eFAST are close, and it is also so for Morris and LH-OAT, RSA and PAWN, which indicates that the methods of the same category have similar results. This is due to the reason that both Sobol and FAST are variance-based methods, they all calculate the contribution of the variance to the output. Both Morris and LH-OAT compute the first-order partial derivatives of the output. Similarly, RSA and PAWN use empirical CDFs and KS statistics to quantify the sensitivity. These groups of methods share the same principle;

4) comparatively, the results of RSA and PAWN are always quite different from other methods. There may be two reasons: firstly, the generation of empirical CDFs may be inaccurate; secondly, the use of KS statistics to compute Sensitivity Index in both methods may bring instability into the results (sensitivity to sampling) because KS statistics takes into account only the maximum difference between CDFs;

5) ranking of parameters for the three models by different SA methods has many differences, but they are quite close in identifying sensitive and insensitive parameters, which means they are effective in screening.

In general, it can be said that all six methods are effective in computing SI. The results of RSA and (to a smaller extent) PAWN are to be treated with care because they use the (sensitive) KS statistics based only on the maximum difference in CDFs between the behavioural and non-behavioural models' sets.

**4.3 Efficiency**

Figure 8 demonstrates the results of six SA methods for the three models for different number of runs. The minimum number of runs needed to get stable ranking of the parameters can be found in Table 5. As can be seen, among all the methods, Morris and LH-OAT converge quickly and are very stable across all numbers of run: they can get reliable SI and ranking at a very small number of runs (100 base sample for each model for both methods). eFAST is also quite stable, and it can get reliable results after approximately 300 base sample size. Comparing eFAST with Sobol method, it can be concluded that eFAST is more stable and more reliable (note also that at some point Sobol method results even in negative SI). RSA and PAWN are

not very efficient, for the reason already stated in the previous section. RSA performs better than PAWN, especially for GR4J and Hymod model, for it can get stable SI at early runs. For PAWN method, the minimum number of runs to obtain reliable results is larger than for the other methods, and the reason is that it needs sufficiently large number of samples to create smooth eCDFs. Besides, the sample size of the conditioning values will affect the conditional eCDFs. It also needs sufficient number

5    of samples of conditioning values to cover the factor space well: this results in high computational cost since for each conditioning value k*Nc runs of the model are needed.

For all methods, it can be seen that with the increase of the model complexity and number of parameters, the results of SA become less stable. Especially for HBV model, except MAXBAS, all other parameters seem to be of similar sensitivity, therefore there is a considerable fluctuation in results. This can also be seen in GR4J and Hymod in which parameters have

10    similar sensitivity (X1 and X3 for GR4J, SM and BETA for Hymod).

**4.4 Convergence**

Figure 9 presents the estimates of the mean and the 95% Confidence Interval of all SA methods for three models with different number of runs. Overall, with increasing number of runs, the width of CI become narrower and have less and less variation. There are still differences in the width of CI and speed of convergence between the methods. It can be seen that Morris, LH-

15    OAT and RSA converge well already at early runs, and the width of CI are quite narrow across all runs. PAWN method converges comparatively slower and the width of CI is also wider. Sobol method is slowest, especially at small number of runs. The upper and lower bound of SI significantly exceed the range 0 to 1, which is quite unacceptable.

For all methods, similar conclusions as in efficiency can be drawn that with the increase of the model complexity and number of parameters, the uncertainty of SA also goes up. This increase of uncertainty also results in unstable results when

20    the sensitivities of the parameters are close as shown in results of efficiency.

From the results shown above, it is proven that all six methods are effective in calculating Sensitivity Indices, screening and ranking. Their efficiencies, however, differ. The minimum number of runs for computing Sensitivity Indices, ranking and reaching convergence with each method are presented in Table 5.

In general, it takes many more runs to reach convergence, but many less runs is sufficient to obtain reliable ranking of the

25    parameters. Sobol method requires large number of runs to be stable and reach convergence, which is very inefficient. Same as variance-based method, eFAST method is much more efficient and stable. It is a good alternative for Sobol method with high efficiency. Morris and LH-OAT are also quite efficient and can provide results of ranking after relatively small number of runs. Also, the uncertainties of the values of Sensitivity Indices are not so high, and especially they are good at ranking and screening. The density-based methods, however, need sufficient number of runs to produce reliable eCDFs, thus the efficiency

30    is not so high. Furthermore, using KS statistics to compute Sensitivity Indices may be problematic for some types of distributions. Comparing RSA with PAWN, one can see that RSA performs better, especially for ranking, however, due to its design, it provides less detailed analysis of sensitivity.

[Figure]

**5 Discussion and recommendations**

**5.1 Principles behind SA methods**

From the results above we can see that different methods show different performance in computing SI, ranking and convergence. One of the reason is that there are different theories, concepts and principles behind each method, and methods

5  of the same category (sharing similar principles) show similar results. Comparing their performance within each category, it can be seen that GLS methods have the highest efficiency and fastest convergence speed. Variance-based and density-based methods perform less well. GLS methods use first-moments to compute SI and the principles they use are relatively simple. Therefore, the propagation of the uncertainty in the SA is also simpler and more direct. Variance-based methods, however, with much more complex principles, result in higher uncertainty in SA. On the other hand, density-based methods may suffer

10  from the necessity to producing reliable and accurate eCDFs, and the fact that using K-S statistics to compute SI. As a result, they are highly unstable and uncertain. RSA performs better than PAWN, owing to its relatively idea (dividing the factor vectors into only two or several sets).

Although variance-based methods seem to be less efficient in computational cost, they use more sophisticated mathematical and statistical apparatus and quantify sensitivity most accurately. Comparatively, GLS methods use only the first

15  derivatives to compute SI, which is of course carries less information (and we can say, less accurate). Density-based methods are moment-independent, they do not need complex equations or computation to get SI, but their strength in quantifying sensitivity is problematic, as stated earlier. Generally speaking, the efficiency and depth of quantification are in inverse relationship. To obtain greater degree of quantification, it takes more model runs, and aiming to reach higher efficiency will lead to inevitable sacrifices in accuracy and reliability of the results. The method that best balances these two aspects seems to

20  be the eFAST method. It uses variance to quantify the sensitivities, and at the same time, requires much smaller number of runs than Sobol method.

Another aspect to be mentioned is the easiness of methods implementation and their integration with (hydrological) models. If the method is too difficult to implement and integrate with the already existing and operational models, its use may be quite limited. This is especially true for distributed models, when sampling may be required at every grid cell, so it is not

25  realistic to use too complex sampling methods, such as in eFAST. In these situations, methods with very simple principals like RSA and LH-OAT are more suitable.

Density-based methods seem attractive due to their simplicity, but they have two problems. On the one hand, the reliability of eCDF produced is questionable. On the other hand, the use of Kolomogorov-Smirnov statistic to compute the Sensitivity Indices, unstable by design, may lead to slow convergence. However, they have two advantages: first, they are moment

30  independent methods, which do not need complicated computational process; second, the results of SA can be expressed in graphs which provide yet another instrument for analysis. One of the idea that can be explored is to quantify the results not by

[Figure]

[Figure]

K-S statistics, which is the maximum difference between the two eCDFs, but to consider an integral difference (the area between two CDFs).

**5.2 Recommendations for choosing SA methods**

Based on the experiments and considerations presented above, we can formulate the following recommendations for choosing

5   SA method(s):

- For simple conceptual hydrological models (not requiring much time for running them multiple times), variance-based methods as Sobol and eFAST are recommended, because they have a strong theoretical background and provide more insight into sensitivity.
- For more complex hydrological or hydraulic models, that need considerable time to run, GLS methods can be used, since
10  they are more efficient.
- For distributed models, methods with simple concepts and sampling techniques are more suitable, such as RSA and LH-OAT.
- For very complex models, e.g. 2D (or even 3D) models, like flood inundation models, or high resolution groundwater models of large aquifers, the Local SA instead of Global SA can be used (Hill and Tiedeman, 2007), or LSA at a selected
15  limited number of points in the factor (sub)space, for a reduced number of factors.
- In situations when only relative sensitivity of the factors is needed, rather than the exact value of SI, it is advisable to aim only at determining ranking or screening of SA which needs significantly less time than the calculation of global SI.
- If time allows, it is recommended, however, to employ several different SA methods rather than using only one method.

**5.3 Practical framework**

20  Based on the analysis of effectiveness, efficiency and convergence of methods, and the recommendations above, we may suggest a practical framework for Sensitivity Analysis and Uncertainty Analysis, as shown in Fig. 10. We consider both SA and UA to be important phases of model analysis, both focusing on certain aspects of model uncertainty, so it is reasonable to bring them together under one framework.

This framework assumes the model is already calibrated, however, it is also applicable to uncalibrated models for
25  choosing a (limited) set of the (sensitive) parameters to calibrate which can improve the efficiency of calibration process.
In case there is a possibility to employ several methods, we can suggest to select one method from each category: variance-based methods, methods aggregating the local sensitivity measures, and density-based ones; the overall judgement about sensitivity will be then better informed. If time does not allow for a large number of runs, Local Sensitivity Analysis method can also be used for the calibrated or observed values of the factors.

30   It is also recommended to first start with a small number of sample size, and then gradually increasing the sample size until the Sensitivity Indices or ranking converges or stabilizes. The stopping criteria is subjective, depending on one's requirement of accuracy or limitation on number of runs. Note that one should balance between the accuracy of the results and the efficiency to obtain these results.

[Figure]

[Figure]

**5.4 Limitations**

We see the following main limitations of this work:

First, the models used in this study are only conceptual rainfall-runoff models with similar structures, so the results may be different for other types of models.

5 Second, the evaluations we did to SA methods are still qualitative, so to evaluate each aspect of SA methods some more rigorous quantitative standard should be set. For example, when evaluating convergence, a threshold of the CI width should be defined for reaching convergence. Quantitative assessment will strengthen the conclusions of the comparisons.

**6 Conclusions**

SA and UA are important steps for better understanding and evaluation of hydrological models. For complex hydrological
10 models, sample-based SA methods are often used. In this study, six different Global Sensitivity Analysis methods: Sobol, FAST, Morris, LH-OAT, RSA and PAWN are tested on the three conceptual rainfall-runoff models: GR4J, Hymod and HBV with increasing complexity and the number of parameters. The methods are compared according to the three criteria: effectiveness, efficiency and convergence.

The results of each method are not exactly identical, but still similar to each other. All of the methods are proven to be
15 effective. Methods from the same category show similar results as they are based on similar principles. The credibility of density-based methods is slightly undermined for two reasons: first, the reliability of eCDF produced may not be always high; second, the use of Kolomogorov-Smirnov statistic to compute the Sensitivity Indices lead to slow convergence.

The evaluation of each method's efficiency demonstrates that GLS methods as Morris and LH-OAT are very efficient and stable in computing SI and ranking. Sobol method can provide quantitative results of SA, but it requires large number of
20 runs to obtain stable results. eFAST is much more stable and efficient than Sobol, thus it may be seen as a good alternative for Sobol method. The efficiency of density-based methods is not so high, but RSA can give reliable results of ranking with small number of runs.

All the methods need significant number of runs (>8000) to reach convergence. The uncertainty in the values of Sensitivity Indices is not negligible. One should be careful when interpreting the results if the number of samples is not sufficiently large.
25 The difference in efficiency of different methods may be due to the difference in the underlying principles. Methods based on simple concepts are more efficient and stable. Methods based on the more complex concept seem to be less stable and efficient, however, their quantification of sensitivity is more accurate and reliable.

The presented recommendation for choosing SA methods, and the framework for SA and UA based on effectiveness, efficiency and convergence, as well as ease of integration with the models, add to other useful SA frameworks (workflows)
30 (e.g. Pianosi et al., 2016), and may be of assistance for practitioners assessing reliability of their models.

[Figure]

[Figure]

Future work will be aimed at considering more SA methods (the first candidate being VARS, Razavi and Gupta, 2016a, 2016b), developing quantitative and more informed measures for their assessment, and testing the results and recommendations against other types of models and scenarios of their practical use.

**Data Availability**

5  Data has been made available by the Department of Hydrology and Meteorology (Nepal) for Masters research at IHE Delft, for which the authors express their gratitude. With questions about the data availability the interested readers may want to contact this organisation directly (http://www.dhm.gov.np/).

**Acknowledgements**

The authors are grateful to the Dutch Ministry of Infrastructure and Environment for providing financial support for the first
10  author when she followed the Master programme in Water Science and Engineering (specialisation in Hydroinformatics) at the IHE Delft Institute for Water Education. The authors also acknowledge support of the grant No. 17-77-30006 of the Russian Science Foundation, and of the Hydroinformatics research fund of IHE Delft.

**Table 1.** Description and ranges of parameters in GR4J mdoel.

| Parameter | Description | unit | Lower bound | Upper bound |
|:---:|:---:|:---:|:---:|:---:|
| $X_1$ | Production store: Storage of rainfall in the surface of soil | mm | 1 | 1500 |
| $X_2$ | Groundwater exchange coefficient: a function of groundwater exchange which influence routing store | mm | -10 | 5 |
| $X_3$ | Routing storage: amount of water can be stored in soil porous | mm | 1 | 500 |
| $X_4$ | Time peak: the time when the ordinate peak of flood hydrograph is created | day | 0.5 | 4 |

[Figure]

[Figure]

**Table 2.** Description and ranges of parameters in Hymod mdoel.

| Parameter | Description | unit | Lower bound | Upper bound |
|:---:|:---:|:---:|:---:|:---:|
| SM | Maximum soil moisture | mm | 0 | 400 |
| BETA | Exponential parameter in soil routing | - | 0 | 2 |
| ALFA | Partitioning factor | - | 0 | 1 |
| RS | Slow reservoir outflow coefficient | - | 0 | 0.1 |
| RF | Fast reservoir outflow coefficient | - | 0.1 | 1 |

[Figure]

**Table 3.** description and ranges of parameters in HBV model.

| Parameter | Description | Unit | Lower Bound | Upper Bound |
|:---:|:---:|:---:|:---:|:---:|
| FC | Maximum soil moisture content | mm | 50 | 500 |
| LP | Limit for potential evapotranspiration | - | 0.3 | 1 |
| ALFA | Response box parameter | - | 0 | 4 |
| BETA | Exponential parameter in soil moisture | - | 1 | 6 |
| K | Recession coefficient for upper tank | mm/d | 0.05 | 0.5 |
| K4 | Recession coefficient for lower tank | mm/d | 0.01 | 0.3 |
| PERC | Percolation from upper to lower tank | mm/d | 0 | 8 |
| CFLUX | Maximum value of capillary flow | mm/d | 0 | 1 |
| MAXBAS | Transfer function parameter | d | 1 | 3 |

[Figure]

**Table 4.** Experimental set-up for evaluation of SA methods.

| Method | Measure | Sampling method | Required number of runs | Parameters within the method | Benchmark run | Number of base samples for evaluation |
|---|---|---|---|---|---|---|
| Sobol | Sobol total-order index | LHS | $(k+2) \times N$ | - | $N=10000$ | $N$ =100/200/300/500/1000/2000/3000/5000 |
| eFAST | FAST total-order index | FAST sampling | $k \times N$ | $M_s = 4$ $N_{cs} = 1$ | $N=10000$ | $N$ =100/200/300/500/1000/2000/3000/5000 |
| Morris | Modified mean of Effect Elementary | Morris one at a time | $(k+1) \times N$ | $p = 32$ $\Delta = 0.5161$ | $N=10000$ | $N$ =100/200/300/500/1000/2000/3000/5000 |
| LH-OAT | Effect S | LHS | $(k+1) \times N$ | $\Delta = 0.05$ | $N=10000$ | $N$ =100/200/300/500/1000/2000/3000/5000 |
| RSA | Mean of KS statistics | LHS | $N$ | - | $N =k \times 10000$ | $N$ =100/200/300/500/1000/2000/3000/5000 |
| PAWN | Max of KS statistics | LHS | $N_u+k \times n \times N_c$ | - | $N_u =500$ $n =40$ $N_c=250$ | $[N_u, n, N_c] =$ [30,10,10]/[50,10,20]/[100,15,20]/[100,20,25]/[200,25,40]/[200,25,80]/[200,30,100]/[500,50,100] |

Notes:

$k$ is the number of parameters; $N$ is the base sample size, $M_s$ is the number of higher harmonics to be considered; $N_{cs}$ is the number of search curves; $N_u$ is the number of samples for constructing unconditional CDFs; $n$ is the number of conditioning values for each parameter, $N_c$ is the number of samples for constructing conditional CDFs. For the detailed explanation of the parameters within each method please refer to the literature referred in Sect. 2.

[Figure]

[Figure]

**Table 5.** Minimum number of runs for computing Sensitivity Indices, ranking and reaching convergence for different SA method.

| Method | Minimum number of run for GR4J | | | Minimum number of run for Hymod | | | Minimum number of run for HBV | | |
|--------|------|---------|------------------|------|---------|------------------|------|---------|------------------|
| | SI | Ranking | Conver-gence | SI | Ranking | Conver-gence | SI | Ranking | Conver-gence |
| Sobol | 15000 | 6000 | 60000 | 35000 | 2100 | 70000 | 11000 | 22000 | 110000 |
| eFAST | 1188 | 388 | - | 2485 | 485 | - | 8937 | 8937 | - |
| Morris | 1000 | 500 | 10000 | 1200 | 600 | 18000 | 5000 | 2000 | 20000 |
| LH-OAT | 1000 | 500 | 10000 | 1200 | 600 | 18000 | 5000 | 5000 | 20000 |
| RSA | 4000 | 400 | 8000 | 2500 | 500 | 15000 | 10000 | 10000 | 30000 |
| PAWN | 12200 | 8200 | 40500 | 15200 | 10200 | 50500 | 18200 | 9200 | 100500 |

[Figure]

[Figure]

[Figure]

**Fig. 1. Graphical expression of Sensitivity Analysis.**

[Figure]

[Figure]

[Figure]

**Fig. 2. Location of the Bagmati catchment. Triangles denote the rainfall stations and circles denote the discharge gauging stations (Shrestha, 2009).**

[Figure]

[Figure]

[Figure]

**Fig. 3. Hydrograph of the Bagmati Catchment from 1 March 1991 to 31 December 1995.**

[Figure]

[Figure]

[Figure]

**Fig. 4. Scatter plot of RMSE against parameter values with 10000 runs for GR4J model.**

[Figure]

[Figure]

[Figure]

**Fig. 5. Scatter plot of RMSE against parameter values with 10000 runs for Hymod model.**

[Figure]

[Figure]

**Fig. 6. Scatter plot of RMSE against parameter values with 10000 runs for HBV model.**

[Figure]

[Figure]

**Fig. 7. Sensitivity Indices (normalised) of six SA methods with benchmark run for GR4J (a), Hymod (b) and HBV (c) model, the number in the grid indicates the rank of the parameter within each SA method.**

[Figure]

[Figure]

**Fig. 8. Sensitivity Indices of different SA methods with different number of runs for GR4J (left column), Hymod (mid column) and HBV (right column) model, the horizontal axis is in log scale.**

[Figure]

[Figure]

**Fig. 9. Estimate of mean and 95% CI of different SA methods with different number of runs for GR4J (left column), Hymod (mid column) and HBV (right column) model, the horizontal axis is in log scale.**

[Figure]

[Figure]

[Figure]

**Fig. 10. Framework for Sensitivity Analysis and Uncertainty Analysis of hydrological model.**

---

## Referee Comment (RC2) · W. Becker (Referee) · 15 Mar 2018

This paper presents a comparison of six sensitivity analysis approaches on three hydrological models. The methods in question are the Sobol' method, eFAST, the method of Morris (elementary effects), LH-OAT, RSA, and PAWN. The authors calculate measures of effectiveness (a rough comparison of the results between methods), efficiency (minimum runs required to reach some criteria of "effective results"), and convergence (calculation of confidence intervals using bootstrap).

The paper is reasonably clear, but unfortunately it lacks focus and novelty. To begin with the latter, the paper does not really add anything over previous comparison studies. It compares one reasonably-new approach (PAWN, although this is very similar to more established moment-independent methods), but compares it to well-established methods that have been around for a long time, and subject to many comparisons. Even inside the hydrology domain, there have been many comparisons of sensitivity analysis techniques, as noted by the authors. To add novelty in this respect, the authors would have to look at very recent developments in sensitivity analysis, perhaps including the latest metamodeling techniques, or methods that account for correlations between input parameters, multivariate output, or other more cutting-edge topics in sensitivity analysis.

From the perspective of focus, it is not really clear precisely what the authors are trying to investigate with this paper. Their main conclusions seem to be that all the methods are useful, but one should be careful to ensure that results have converged, and that different methods anyway interpret sensitivity in different ways. This kind of advice can be found in textbooks, so it is not really publishable material. In order to improve the focus of the paper, the authors should consider focusing a bit more: for example, are these results particular to their models, and to what extent can they be generalised? How do their results compare with other comparisons, and why might theirs be different? What is it about their models that makes one method more suitable than another, for example in terms of input distributions, dimensionality, degree of nonlinearity and so on? If the paper could give some kind of more in-depth analysis of why certain methods perform better than others, that would already help. However as it is, there is really nothing that a reader can take away from the paper that cannot be found in many other places.

To summarise, to be novel, a comparison study should either study very recent methods that have not been subject to comparisons so far, or (and) go into a level of depth that uncovers new conclusions about the methods in question. While the authors made an attempt to look at different aspects performance, the conclusions they draw show that no real novelty has been produced here. Therefore I must recommend a rejection.

I would encourage the authors to think carefully about the added value of their paper to other researchers, and use that as guide for how to improve the paper, perhaps by incorporating more advanced techniques or really drilling down to the differences between the methods in considerable detail.

---

## Referee Comment (RC3) · T. Wagener (Referee) · 24 Mar 2018

Providing insight and guidance for users of Global Sensitivity Analysis (GSA) in selecting the appropriate method for their situations is a very current and relevant issue. One thing that has certainly not been sufficiently assessed is how we can combine different approaches in a multi-method approach to GSA (e.g. discussion in Pianosi et al., 2016, EM&S). So, I think that there is some value to the work done here. There is also clearly some more work to be done by there authors as the other reviewers already mentioned and I will not discuss the same again here. Rather I am making some more suggestions for the authors to consider.

[1] Reduce the content and more clearly focus the study. Certainly, I would avoid presenting a framework. I think that comparing methods in detail and discussing how they can and should be combined is much more valuable. We know they give different results, but how can we use this? See point [4] as well.

[2] Some of the conclusions in model selection are trivial or are not consistent with example studies already in the literature. For example, there are already quite a few GSA studies using variance-based approaches with distributed hydrological or environmental models (in contrast to the authors' third recommendation) (e.g. van Werkhoven et al., 2008, GRL).

[3] If the focus lies on convergence of these algorithms, then you should really assess this issue in great detail and study for example whether convergence depends on the catchment studies or other things that can be varied between model runs (such as different uncertainties in the input and output data).

[4] Figure 10 is a generic flowchart for GSA and as such more suitable for a review paper or a book chapter. I do not see how this advances on past work and would take it out. Focusing on what you can learn from applying these different methods would be much more valuable and interesting.

---

## Editor Comment (EC1) · N. Romano (Editor) · 25 Mar 2018

Dear Authors, With a view of the stimulating comments and constructive criticisms received so far, I would suggest you should start providing some preliminary responses so as to feed this discussion phase on your manuscript.

---

## Author Comment (AC1) · 28 Mar 2018

(Please see the attached doc file)

MAIN REVIEW (GENERAL COMMENT)

I am afraid I gave up on this paper (after making quite a lot of comments in the manuscript) at the point where Figures 4-6 are introduced and demonstrate the irrelevance of sensitivity analysis in the chosen case study. Almost certainly in these cases the performance of the model has more to do with uncertainty in the input and output data, that is totally neglected, than the factors included in the analysis. This

is indicative of the apparently naïve way the issues associated with sensitivity analysis are presented in the introductory sections which can only be described as poorly presented. In particular, there is no real recognition of the potential for complexity of surfaces with sometimes rapidly changing covariation, including changes of sign, of factors in producing the outputs (that can be concealed in plots such as Fig 4) - yet such behaviour is common for real model applications. Also, despite the discussions of the last 30 years, the authors still seem (surprisingly?) to believe in the possibility of an optimum calibrated model.

REPLY. Indeed the authors have been trying to follow a quite widely accepted idea that SA (of parameters) has a value. We also agree that model performance often depends on the data uncertainty than on the parametric uncertainty, however studying this relationship was the the objective of this paper. We agree that taking into account covariation as well would be the right thing to do. And yes, we believe in the possibility of an optimum calibrated model, as most people who do "SA of a calibrated model". We appreciate the need to assume multi-model representation of reality, the equifinality principle, and we have even contributed (modestly) to developing multi-model approaches and UA - but our experience with practitioners is that they typically want to use a single deterministic model rather work with multiple ones. We have to continue explaining them that assumption of a single optimal model could be misleading, and importance of explicit account of uncertainty.

The authors recognise that nearly all past intercomparisons of SA methods have suggested that different methods give different results, and that the same method might give different results when used with different outputs. So it is here too. This is not therefore unexpected, so where is the value in this paper, or in continuing to explore further SA methods as they suggest.

REPLY. Indeed, this study confirms what has been demonstrated earlier, and in this respect its value is limited. The main idea here was to present "practical experience" (see the title) and to have a multiplicative effect by several models, several methods,

several case studies. It seems though it was not enough.

Are the results really ever used to decide parameters "on which more resources can be put to ensure their higher accuracy". How would you actually do this for the conceptual models used in the paper, when it is effective values of model parameters that are needed to give good predictions? That would be a much more interesting paper.

REPLY. Agreed, the statement "on which [parameters] more resources can be put to ensure their higher accuracy" in relation to conceptual models is wrong. It seems though it was not enough.

As it is I cannot suggest that this paper is suitable for publication.

Keith Beven

REPLY. Clear; we accept this. Our intention was to present the "Practical experience and framework", and we realise (especially taking into account the other referees' comments) that in the present form the study cannot be considered as a "research paper".

Thank you very much for the attention to this paper, and for the comprehensive review.

- - - - - - - - - - - - - -

THE FURTHER TWENTY FIVE SPECIFIC COMMENTS MADE DIRECTLY IN THE MANUSCRIPT BY THE REFEREE

Comment #1 Page 2 Line 1 There are various definitions of UA proposed by different researchers (Cacuci, 2005; Pappenberger and Beven, 2006; Tong, 2006; Saltelli et al, 2008; Bastin et al, 2013). UA gives a qualitative or quantitative assessment of the uncertainty in the model results. The results can be qualitatively expressed in a graph showing the spread an ensemble of values or a distribution, as probabilistic flood maps, etc Comment: This paper is not about UA?

REPLY. We see SA as part of UA.

Comment #2 Page 2 Line 5 Due to the complexity and non-linear nature of hydrological models, it is hard to use analytical methods to study the uncertainty of hydrological models Comment: But you can use semi-analytical methods for local SA.

REPLY. Yes, indeed, but we had limited resources to cover all known methods.

Comment #3 Page 2 Line 6 Therefore, non-intrusive, sampling-based methods are commonly used, generally referred to as Monte Carlo Simulation (MCS), which can be seen as the simulation of a system that encloses stochastic or uncertain components. Comment: What does this mean in this context? And is not the point that MCS is only one way of investigating the output response surface.

REPLY. The only thing we wanted to say that MC is that sampling is an alternative to analytical methods.

Comments #4 Page 2 Line 15 "how the uncertainty in the output of a model (numerical or otherwise) can be apportioned to different sources of uncertainty in the model input" (Saltelli et al., 2008). Comment: No! Because you cannot actually apportion for a complex response surface - that requires additional (and not necessarily realistic) assumptions such as linear variance decomposition

REPLY. We agree, in general case it is so. However referring to "different sources of uncertainty in the model input" (Saltelli et al., 2008) we did not say "separately".

Comments #5 Page 2 Line 17 (One may notice that this definition is not comprehensive, since uncertainty not only comes from model inputs but also from parameters, so for this reason, we will use the term "factor" instead of "model input".) Comment: But Saltelli et al. include parameters as "inputs". "We will use"??? Saltelli et al already use factors.

REPLY. Agreed, we could have referred to Saltelli et al. (Reason for explicit mentioning this is that in many papers it is not always clear if "inputs" include "parameters" or not.)

Comments #6 Page 2 Line 19 The main aim here is to identify the degree with which

changes in various factors (manifesting the corresponding uncertainty) influence a change in model output. Comment: Is this really correct? - for a SA you do not need to specify any prior on the factors, so you are not actually taking account of that uncertainty, only looking at outputs in response to variation in the inputs.

REPLY. What we wanted say is just that "changes" manifest "uncertainty" without saying that uncertainty is expressed probabilistically (or one can say we assume a discrete distribution).

Comment #7 Page 2 Line 26 On the other hand, for distributed hydrological models, whose parameters can be gathered from the field, SA can help to target the most important parameters, on which more resources can be put to ensure their higher accuracy. Comment: This is standard fare – but is it correct? How would you do this when what is needed is effective value of the parameters? And how do you know that other parameters are not important when response surfaces are complex?

REPLY. We agree this statement does not fully reflect the mentioned complexity.

Comment #8 Page 2 Line 28 it is not yet really known what is the optimal parameter vector, and hence it is possible that sensitivity is investigated considering non-feasible parameters values. Comment: But THERE IS NO OPTIMAL PARAMETER VECTOR – that depends on both data period and evaluation measure or measures. That discussion has gone for 30 years!!! If you believe in an optimum then why not just evaluate sensitivity around that optimum????

REPLY. We appreciate this comment, but presented our opinion in answering the "general comment" above. (Perhaps using the term "optimal" is not very fortunate.)

Comment #9 Page 2 Line 31 SA is conducive to UA, and the main difference between their aims lies in that SA tries to explicitly apportion the uncertainty of the output to the different factors. Comment: See earlier comment – you cannot be sure that this is correct for any complex surface.

REPLY. Agreed. To be revised.

Comment #10 Page 3 Line 1 whereas UA provides a more general and often more detailed and rigorous account of model uncertainty. Comment: In what sense – can surely be based on exactly the same samples?

REPLY. We would like to stay with this (quite general) statement. Discussing SA vs UA further would pull us away from the main theme of the paper.

Comment #11 Page 3 Line 8 SA is typically categorized into Local Sensitivity Analysis (LSA) and Global Sensitivity Analysis (GSA). Comment: Surely needs to come before any mention of MCS.

REPLY. Agreed. To be corrected.

Comment #12 Page 3 Line 9 LSA concentrates on the sensitivity of factors at particular points in the factor space, for example, around the vector of the calibrated parameters. Comment: See earlier comment.

REPLY. Agreed.

Comment #13 Page 3 Line 12 A simplest expression of local sensitivity is the first-order partial derivatives of output to the factors. Define a model y = f(x), where y is the output of the model; x is factor of the model. The sensitivity of the factor (S) is defined as: ðÍŚĘðÍŚŰ = ΔðÍŚęðÍŚŰ ΔðÍŚěðÍŚŰ (1) 15 where i is the i-th factor of the model. (Note, that in quite many studies instead of model output y the model error is used, e.g. Root Mean Squared Error or Mean Absolute Error.) Higher value of Si indicates higher sensitivity of the factor. Such measure of sensitivity is often called Sensitivity Index (SI). Figure 1 shows the expression of sensitivity of a model with two parameters (factors). Comment: You do not use this so why does it need repeating here.

REPLY. Agreed. To be corrected.

Comment #14 Page 3 Line 19 If we randomly sample several points in the whole parameter space, and obtain Si for each sample point. Comment: Oh come on!! So if you average large positive and negative values you will get zero. Are you really so unaware of the issue in SA?

REPLY. Agreed. This is a text-book issue which should have been mentioned.

Comment #15 Page 3 Line 28 It is demonstrated that different methods result in different ranking of factors, thus solid conclusions about the sensitivity of the factors are impossible to draw. Comment: Ok so not completely unaware – but what do these studies imply for your study??

REPLY. We are aware of the fact that solid conclusions about SA results may be hard to draw. Instead, we give suggestions rather than conclusions about how to choose appropriate SA methods based on several aspects (effectiveness, efficiency, convergence, implementation, etc.).

Comment #16 Page 3 Line 30 The result demonstrates qualitative SA methods are more efficient than quantitative SA methods, whereas quantitative SA methods are more robust and accurate. Comment: You need to at least say what is involved in a qualitative SA?

REPLY. Agreed. To be corrected.

Comment #17 Page 5 Line 20 They are sorted from best to worst, in which the first group produces the best 10% results (e.g. the results with least 10% model error), the second group produces the best 10%-20% results and so on. Comment: Can also be from largest to lowest for any output variable.

REPLY. Agreed. To be formulated better.

Comment #18 Page 6 Line 26 However, Razavi and Gupta (2015) have pointed out that they may suffer from scale issue, that is, the selection of the step size may influence the results due to the complexity of response surface of the model. Comment: Indeed!! See earlier comment.

REPLY. Agreed, to be addressed.

Comment #19 Page 7 Line 17 The advantage of meta-modelling is that by simplification of the original complex model, the overall running time is considerably decreased; the trade-off is a possible loss of accuracy. Comment: No. none of these simplify the complex model – they only interpolate the output response surface between the known values (with or without uncertainty). In doing so they might get quite the wrong local sensitivities since each method is constraining the gradient in some way. Reply:

REPLY. Agreed. Had to be formulated better.

Comment #20 Page 7 Line 23 Different SA methods have different concepts and principles behind them, and, accordingly, the Sensitivity Indices may have different meaning and metrics. Comment: So why are you comparing them rather than accepting that they might produce different outcomes?

REPLY. Please see comment #15.

Comment #21 Page 7 Line 25 The evaluation of SA methods' effectiveness is aimed at finding out whether the relative Sensitivity Indices, ranking and screening of parameters have sense and indeed can be used in SA. Comment: But how do you know when you make no evaluation of the real nature of the surface in a complex case?

REPLY. Good point; a better explanation required. Of course we run the models for 10000 times in the first place. But maybe the provided analysis of the results is not deep enough.

Comment #22 Page 8 Line 5 Unlike assessing the accuracy of a 5 hydrological model, which can be compared with the observation values, for sensitivity there are no 'observations' to be compared with. Comment: But even then your observations may be significantly uncertain

REPLY. True; this statement requires will be given better explanation.

Commnet #23 Page 8 Line 6 h. To start somewhere, we will initially randomly sample a large number (say, 10,000) parameter (factor) vectors and run the model for each of them. Comment: That is not large for more than 4 or 5 factors? Reply:

REPLY. Indeed - but we have start somewhere...

Comment #24 Page 8 Line 7 The RMSE of the model output will be plotted against parameter values as a scatter plot which will provide a rough image of the sensitivity of each parameter. Comment: RMSE? – but you have just said you have no observations to compare against? Since this is your reference it needs to be explained much more clearly.

REPLY. Indeed, perhaps not very well formulated. Here "observation" means there is no observation for sensitivity itself, not the model output. Though observations are highly uncertain in some sense, still they are the best references for evaluating model output. But when doing SA, you don't have such "observations" as references.

Comment #25 Page 10 Line 20 The model was run 10,000 times; the scatter plots of the ðİŘỳðİŠĚðİŚĂðİŚĘðİŘỳ against parameters for the three models are shown in Fig. 4-6. Comment: So given these figures why is SA relevant at all?

REPLY. The reason why we run the models 10000 times is to have a preliminary assessment of the model response and thus to draw the conclusions about the parameter sensitivity as a references for evaluation. This methodology have been reported in previous study as in Wagener et al., 2001, Hall et al., 2009, Pianosi and Wagener, 2015, etc. We will provide more explanation.

We would like to thank Professor Beven for the attention given to this paper and the review, and pointing out the deficiencies, unclarities, and providing valuable suggestions.

Please also note the supplement to this comment:

https://www.hydrol-earth-syst-sci-discuss.net/hess-2018-78/hess-2018-78-AC1-supplement.pdf

---

## Author Comment (AC2) · 28 Mar 2018

Replies to the interactive comment by W. Becker (Referee)

RC: This paper presents a comparison of six sensitivity analysis approaches on three hydrological models. The methods in question are the Sobol' method, eFAST, the methodof Morris (elementary effects), LH-OAT, RSA, and PAWN. The authors calculate measures of effectiveness (a rough comparison of the results between methods), efficiency (minimum runs required to reach some criteria of "effective results"), and convergence (calculation of confidence intervals using bootstrap).

The paper is reasonably clear, but unfortunately it lacks focus and novelty. To begin with the latter, the paper does not really add anything over previous comparison studies. It compares one reasonably-new approach (PAWN, although this is very similar to more established moment-independent methods), but compares it to well-established methods that have been around for a long time, and subject to many comparisons. Even inside the hydrology domain, there have been many comparisons of sensitivity analysis techniques, as noted by the authors. To add novelty in this respect, the authors would have to look at very recent developments in sensitivity analysis, perhaps including the latest metamodeling techniques, or methods that account for correlations between input parameters, multivariate output, or other more cutting-edge topics in sensitivity analysis.

REPLY. Indeed, we agree the paper does not go beyond the comparison methods employed earlier. As it is seen from the title, the purpose was however to present the "practical experience and framework", based on several examples and methods which could be useful for practitioners. We agree we could have gone further and cover the mentioned areas, but the scope of this study was limited, mainly by time, and it is planned to extend the scope.

RC: From the perspective of focus, it is not really clear precisely what the authors are trying to investigate with this paper. Their main conclusions seem to be that all the methods are useful, but one should be careful to ensure that results have converged, and that different methods anyway interpret sensitivity in different ways. This kind of advice can be found in textbooks, so it is not really publishable material. In order to improve the focus of the paper, the authors should consider focusing a bit more: for example, are these results particular to their models, and to what extent can they be generalised? How do their results compare with other comparisons, and why might theirs be different? What is it about their models that makes one method more suitable than another, for example in terms of input distributions, dimensionality, degree of non-linearity and so on? If the paper could give some kind of more in-depth analysis of why

certain methods perform better than others, that would already help. However as it is, there is really nothing that a reader can take away from the paper that cannot be found in many other places.

REPLY. We appreciate these comments. Again, intention was to orient practitioners and to systematically compare the methods w.r.t. the important criteria (effectiveness, efficiency, convergence, implementation, etc.) on a number of cases and models. We agree the analysis in this study could have been deeper.

RC: To summarise, to be novel, a comparison study should either study very recent methods that have not been subject to comparisons so far, or (and) go into a level of depth that uncovers new conclusions about the methods in question. While the authors made an attempt to look at different aspects performance, the conclusions they draw show that no real novelty has been produced here. Therefore I must recommend a rejection.

REPLY. Accepted. Our intention was, as it can be seen from the title, to present the "practical experience and framework", and to give additional orientation for practitioners, but we realise (also taking into account the other referees' comments) that in the present form the study is not deep enough to be presented as a "research paper" in HESS.

Thank you very much for the attention to this paper, and for the useful comments.

---

## Author Comment (AC3) · 28 Mar 2018

RC: Providing insight and guidance for users of Global Sensitivity Analysis (GSA) in selecting the appropriate method for their situations is a very current and relevant issue. One thing that has certainly not been sufficiently assessed is how we can combine different approaches in a multi-method approach to GSA (e.g. discussion in Pianosi et al., 2016, EM&S). So, I think that there is some value to the work done here. There is also clearly some more work to be done by there authors as the other reviewers already mentioned and I will not discuss the same again here. Rather I am making some more suggestions for the authors to consider.

[Figure]

[1] Reduce the content and more clearly focus the study. Certainly, I would avoid presenting a framework. I think that comparing methods in detail and discussing how they can and should be combined is much more valuable. We know they give different results, but how can we use this? See point [4] as well.

REPLY. Indeed, we have not considered the (powerful) idea of combining of methods presented in Pianosi et al 2016, and it was not our intention in this study. We have done some work in the past contributing to the multi-model approaches in hydrological modelling, so fully share the idea of "multi". Of course methodologically it is a tricky thing, since the methods called SA are doing actually conceptually very different things (but why not - since we do not hesitate to build hybrid environmental models combining different modelling paradigms). We could have included more methods as well, and interpret the results deeper.

[2] Some of the conclusions in model selection are trivial or are not consistent with example studies already in the literature. For example, there are already quite a few GSA studies using variance-based approaches with distributed hydrological or environmental models (in contrast to the authors' third recommendation) (e.g. van Werkhoven et al., 2008, GRL).

REPLY. Indeed, we have not considered the (powerful) idea of combining of methods presented in Pianosi et al 2016, and it was not our intention in this study. It is also methodologically a tricky thing, since the methods called SA are doing actually conceptually very different things. We could have included more methods as well, and interpret the results deeper, to advance beyond a "trivial comparison". Concerning the conclusion 2 (re. use of GSA for distributed models), indeed we could have extended it beyond recommending only RSA and LH-OAT, and going deeper, commenting on the possibilities of using of GSA more generally. We are aware of the mentioned approaches, referring in the paper to the paper Tang, Y., Reed, P., Van Werkhoven, K., and Wagener, T.: Advancing the identification and evaluation of distributed rainfall-runoff models using global sensitivity analysis, WRR., 43, 1–14, 2007, and we are

thankful for the (new for us) reference to van Werkhoven, Wagener, Reed, Tang (2008) Rainfall characteristics define the value of streamflow observations for distributed watershed model identification, GRL 2008. Dealing with distributed models is indeed on our research agenda.

[3] If the focus lies on convergence of these algorithms, then you should really assess this issue in great detail and study for example whether convergence depends on the catchment studies or other things that can be varied between model runs (such as different uncertainties in the input and output data).

REPLY. Indeed, this issue could have been studied in greater detail, and it is planned to do.

[4] Figure 10 is a generic flowchart for GSA and as such more suitable for a review paper or a book chapter. I do not see how this advances on past work and would take it out. Focusing on what you can learn from applying these different methods would be much more valuable and interesting.

REPLY. Indeed, the framework is quite generic, and generally follow a number of frameworks presented earlier. Our intention was, as it can be seen from the title, to present the "practical experience and framework", and to provide additional orientation for practitioners, who could follow the flowchart and evaluate for their cases what methods to use. We aimed to stress explicitly effectiveness, efficiency, convergence, implementation, etc. as the important criteria for comparison but it seems now we were not convincing enough.

We realise (also taking into account the other referees' comments) that in the present form the study is not in the form to be presented as a "research paper" in HESS.

We would like to thank Professor Wagener for the time given to review this paper, pointing out deficiencies, and providing the very valuable suggestions.

We will be evaluating our options concerning extension and deepening of this research

(which requires time and resources), and the target audience of this paper.

---

## Author Comment (AC4) · 28 Mar 2018

Dear Edtior,

We would like to thank all referees and the Editor for the time given to review this paper, pointing out deficiencies, and providing the very valuable suggestions, especially on the ways to deepen the analysis and improve the interpretation of results.

Our intention was, as it can be seen from the title, to present the "practical experience and framework", and to provide additional orientation for practitioners among a variety of methods, who could follow the logical framework/workflow and assess what

methods to use and why. We aimed to stress explicitly effectiveness, efficiency and convergence, as the important criteria for comparison, but it appears we were not convincing enough. We appreciate the recommendations of the referees and accept that in the present form the study is not in the form to be presented as a "research paper" in HESS.

We will be evaluating our options concerning extension and deepening of this research (which requires time and resources), the objectives, positioning, and the target audience of this paper.